# Serum Biomarker Concentrations upon Admission in Acute Traumatic Brain Injury: Associations with TBI Severity, *Toxoplasma gondii* Infection, and Outcome in a Referral Hospital Setting in Cameroon

Franklin Chu Buh [1,2,*] ⓘ, Germain Sotoing Taiwe [1] ⓘ, Firas H. Kobeissy [3] ⓘ, Kevin W. Wang [3] ⓘ, Andrew I. R. Maas [4] ⓘ, Mathieu Motah [5], Basil Kum Meh [1], Eric Youm [6] ⓘ, Peter J. A. Hutchinson [7] and Irene Ule Ngole Sumbele [1]

[1] Department of Animal Biology and Conservation, Faculty of Science, University of Buea, Buea P.O. Box 63, Cameroon; taiwe_sotoing@yahoo.fr (G.S.T.); mehbasil90@gmail.com (B.K.M.); sumbelei@yahoo.co.uk (I.U.N.S.)

[2] Panafrican Hospital Center, Douala P.O. Box 13152, Cameroon

[3] Center for Neurotrauma, Multiomics & Biomarkers (CNMB), Department of Neurobiology, Neuroscience Institute, Morehouse School of Medicine, 720 Westview Dr SW, Atlanta, GA 30310-1458, USA; firasko@gmail.com (F.H.K.); kawangwang17@gmail.com (K.W.W.)

[4] Department of Neurosurgery, Antwerp University Hospital, University of Antwerp, 2000 Edegem, Belgium; andrew.maas@uza.be

[5] Department of Surgery, Faculty of Medicine and Pharmaceutical Sciences, University of Douala, Douala P.O. Box 2701, Cameroon; motmath@yahoo.fr

[6] Holo Healthcare, Nairobi 00400, Kenya; ericprincelds@gmail.com

[7] Department of Clinical Neuroscience, University of Cambridge, Cambridge CB2 0QQ, UK; pjah2@cam.ac.uk

* Correspondence: franklinbuhchu@gmail.com; Tel.: +237-676489649

**Abstract:** Despite the available literature on traumatic brain injury (TBI) biomarkers elsewhere, data are limited or non-existent in sub-Saharan Africa (SSA). The aim of the study was to analyse associations in acute TBI between the admission serum biomarker concentrations and TBI severity, CT-scan findings, and outcome, as well as to explore the influence of concurrent *Toxoplasma gondii* infection. The concentrations of serum biomarkers (GFAP, NFL Tau, UCH-L1, and S100B) were measured and *Toxoplasma gondii* were detected in the samples obtained <24 h post injury. GOSE was used to evaluate the 6-month outcome. All of the biomarker levels increased with the severity of TBI, but this increase was significant only for NFL ($p = 0.01$). The GFAP values significantly increased ($p = 0.026$) in those with an unfavourable outcome. The Tau levels were higher in those who died ($p = 0.017$). GFAP and NFL were sensitive to CT-scan pathology ($p$ values of 0.004 and 0.002, respectively). The S100B levels were higher ($p < 0.001$) in TBI patients seropositive to *Toxoplasma gondii.* In conclusion, NFL was found to be sensitive to TBI severity, while NFL and GFAP were predictive of CT intracranial abnormalities. Increased levels of GFAP and Tau were associated with poorer outcomes 6 months after TBI, and the S100B levels were significantly affected by concurrent *T. gondii* infection in TBI patients compared with the seronegative patients.

**Keywords:** serum biomarkers; traumatic brain injury; CT-scan findings; outcome; *Toxoplasma gondii*

## 1. Introduction

Traumatic brain injury (TBI) remains a major cause of mortality and disability globally [1,2], and more than 50 million people sustain a TBI each year, worldwide [3]. The crude incidence of TBI at the regional level in Europe ranges between 83.3 to 849 per 100,000 people per year [4]. In low- and middle-income countries (LMICs), the incidence of TBI ranges between 150 and 316 cases per 100,000 people [5,6]. In sub-Saharan Africa (SSA), the incidence of TBI is expected to rise to 14 million yearly by 2050 [7–9]. TBI is

extremely common and is associated with complex biological and chemical changes in the brain resulting from the mechanical forces applied to the head [10].

Therefore, understanding the underlying pathomechanisms of TBI and achieving an accurate diagnosis is necessary for effective and patient-oriented treatment [11,12]. Presently, the primary clinical severity indicators for TBI are the Glasgow Coma Scale (GCS), pupil reactivity, and head computed tomography (CT). However, these are limited, as they do not show cerebral pathology at a cellular or molecular level. Hence, several astroglial and neuronal proteins have been proposed as potential biomarkers [13,14].

The importance of biomarkers in traumatic brain injury lies in their capacity to provide insights into injury-induced cellular, biochemical, and molecular changes and to demonstrate the presence of early micro lesions that are difficult to detect using imaging techniques [15]. Protein biomarkers are the most studied, with a focus on S100 calcium-binding protein (S100B), neuron-specific enolase (NSE), glial fibrillary acidic protein (GFAP), ubiquitin C-terminal hydrolase-L1 (UCH-L1), and neurofilament-light (NFL) [3,12,16,17]. In the USA, the Food and Drug Administration (FDA) has approved the UCH-L1/GFAP-based blood test to help determine the need for a CT scan in mild TBI patients [18]. Generally, magnetic resonance imaging (MRI) and blood biomarker measurement are reported to enhance the characterization of injury severity and the type of TBI [19]. Therefore, the prospect of using peripheral blood-based markers synergistically with current clinical diagnostic and prognostic assessments in TBI is attractive for its clinical acceptability, and compared with other invasive procedures, it is cost effective and may quickly and accurately provide specific information about the underlying pathophysiology of TBI, which clinicians need for the formulation of treatment strategies and prognosis [18].

Despite the available literature on TBI biomarkers elsewhere [3,12,20–24], data are limited or non-existent in SSA. In contrast with most developed countries, *T. gondii* is prevalent in SSA, and often involves the central nervous system. Neuro-inflammation is considered an important component of TBI [25–27], and we hypothesized that concomitant *T. gondii* infections might lead to higher biomarker levels after TBI. TBI biomarker research may be particularly promising in resource-limited countries, as the use of biomarkers may rule out the need for costly exams such as CT scans, which is particularly relevant to a setting where a major part of the population lives under poverty. This study aimed to analyse associations in acute TBI between admission serum biomarker concentrations and TBI severity, CT-scan findings, and outcome, and to explore the possible influence of *Toxoplasma gondii* infection.

## 2. Methods

### 2.1. Study Area, Design, and Period

This prospective-cohort study was conducted at the Laquintinie Hospital of Douala (LHD) on acute traumatic brain injury patients. Douala is the economic capital of Cameroon and is a highly cosmopolitan city. It is situated on the south eastern shore of the Wouri river estuary, on the Atlantic Ocean coast, about 230 km west of Yaoundé. It has a wet and a dry season, and has temperatures ranging from 74 °F to 91 °F with an estimated population of about 3.8 million inhabitants [28,29]. LHD is a second-category referral hospital located in the heart of AKWA Douala. It is located in an area of 9 hectares. Its mission is to ensure medical care and quantitative and qualitative medico-sanitary responses in major events such as sporting activities, traffic accidents, disasters, or epidemics. LHD offers many services organized within departments such as the Department of Surgery (paediatric surgery, neurosurgery, ORL, Ophthalmology, Urology, and Orthopaedics/Traumatology, general surgery) and the Department of Emergency, Anaesthesiology, and Reanimation. The neurosurgical department is well established and the hospital has three neurosurgeons, a CT Scanner, and 0.5 tesla MRI. The hospital was chosen for the study because it received the highest number of trauma cases in the Littoral Region of Cameroon, and probably Cameroon at large. The study was conducted over a period of 13 months, from January 2021 to January 2022.

### 2.2. Study Population and Participants

The population of the study consisted of all patients who sustained a traumatic brain injury, mild to severe, and were received at the emergency service of the Laquinitinie Hospital Douala within 24 h of injury during the study period. Demographic details and outcome of this cohort have been reported previously (Buh et al., 2022, submitted). Fifteen healthy individuals were also included in the study as controls. Inclusion criteria: all individuals who were brought to the emergency services of the Laquintinie Hospital Douala who sustained a head trauma and for whom written informed consent was obtained from themselves or their family members, were enrolled in the study. Exclusion criteria: all individuals who were not confirmed to have a TBI (mild, moderate, or severe) by the physician, individuals with neuro-psychiatric problems, thieves, and those patients from whom we could not get blood samples for various reasons.

### 2.3. Sampling Method and Unit

Patients or their families, depending on the severity of TBI and their level of consciousness, were approached to obtain consent to participate in the study. Those who gave their consent were enrolled in the study. Information on the sociodemographic, clinical and injury details, was registered. Blood samples were collected within 24 h after injury for the determination of the serum concentrations of five biomarkers of TBI: glial fibrillary acidic protein (GFAP), ubiquitin C-terminal hydrolase-L1 (UCHL1), total tau protein, neurofilament light (NFL), and calcium-binding protein (S100B). The association of these biomarkers to the severity of TBI (mild, moderate, severe), *T. gondii* infection, CT-scan findings, and 6-month Glasgow outcome Scale-Extended (GOSE) outcome (favourable vs. unfavourable, survival vs. death) were explored. Favourable outcome was interpreted as a GOSE score from 5 to 8, while unfavourable outcome was a GOSE score 1–4. The GOS-E was used to determine the level of recovery post-traumatic brain injury 6 months after injury. The evaluation was done through face-to-face visits or telephone calls. Some (*n* = 8), however, were missed due to contact difficulties after hospital discharge.

### 2.4. Laboratory Methods

Blood samples from the study participants were collected into the dry-tubes (no anti-coagulant added). The samples were kept for 30 min to an hour for coagulation to take place. After this, the samples were centrifuged at 3500 rpm for 15 min to obtain the blood serum. Each serum was aliquoted using micro-pipettes and emptied into two labelled and coded cryo-tubes; two for each sample. Care was taken to ensure the cryotubes had the same code as the corresponding dry tubes. Serum samples were preserved at −80 degrees at the central laboratory of the LHD until shipment. Samples were shipped to the USA (University of Florida, Gainesville, Florida, FL, USA) for analysis under adequate conditions (World courier) and to the University of Buea (Buea, Cameroon).

### 2.5. Measurement of Blood Fluid Biomarkers (GFAP, Tau, UCHL-1, and NFL)

The blood samples were collected within 24 h of injury. GFAP, UCH-L1, t-tau, and NFL were analysed at the University of Florida using single molecule arrays (SiMoA) based 4-plex for research-use-only assay (N4PB, Item number 103345) on the SR-X benchtop assay platform (Quanterix Corp., Lexington, MA, USA). The SiMoA Human Neurology 4-Plex B assay (N4PB) measures four important neurology biomarkers in both cerebrospinal fluid (CSF) and blood. The four targets are neurofilament light (NF-L), total tau, glial fibrillary acidic protein (GFAP), and ubiquitin carboxyl-terminal hydrolase L1 (UCH-L1). The lower limit of detection (LLoD) was as follows: GFAP 1.32 pg/mL, NFL 0.0971 pg/mL, t-tau 0.0236 pg/mL and UCH-L1 0.67 pg/mL. The assay range was as follows: GFAP 0–40,000 pg/mL, NFL 0–2000 pg/mL, UCH-L1 0–40,000 pg/mL, and Tau 0–400 pg/mL. All of the samples, including the controls, were GFAP, NFL, and t-tau and UCH-L1 readout above the LLoD [22,30].

The serum S100B concentration was measured using the Commercial Assay ELISA Kits following the manufacturer's instructions (MyBioSource INC., San Diego, CA, USA, MBS762374, sales@mybiosource.com). One hundred (100) μL of diluted standards, quality controls, dilution buffer (=blank) and samples, were pipetted into the appropriate wells. The plates were incubated at room temperature (37 °C) for 90 min. The wells were washed two times with a wash buffer and the plates were inverted and tapped strongly against a paper towel. After this, 100 μL of Biotin Labelled Antibody solution was added into each well. The plate was incubated at room temperature (37 °C) for 60 min. The wells were washed three times. One hundred (100) μL of Streptavidin−HRP Conjugate was added into each well and incubated at 37 °C for 30 min. The wells were washed two times and 90 μL of TBM Substrate Solution was added into each well. The plates were covered and incubated for 10 min. Fifty (50) μL of stop solution was added to each well and the colour immediately turned yellow, as stated by the manufacturer. The optical density (OD) absorbance was read at 450 nm in a microplate reader immediately after adding the stop solution. The normal serum concentration for S100B in this study was <1 pg/mL.

### 2.6. Measurement of Toxoplasma gondii IgG Antibodies (ELISA)

The seropositivity to *T. gondii* was measured using Commercial Assay ELISA Kits following the manufacturer's instructions (MyBioSource INC, USA, MBS494548, sales@mybiosource.com). One hundred microliters (100 μL) of diluted sera, calibrator, and controls were dispensed into the appropriate wells. For the reagent blank, a 100 μL sample diluent was emptied into the wells. The holder was tapped to remove air bubbles from the liquid and mixed well, and the plate was incubated for 20 min at room temperatur. After this, the liquid from all of the wells was removed and the wells were washed three times with 300 μL of 1X wash buffer. The plate was then blotted on absorbance paper. One hundred microliters of enzyme conjugate were added to each well, which was then incubated for 20 min at room temperature. The enzyme conjugate was removed from all of the wells and they were washed three times with 300 μL of 1X wash buffer; then, 100 μL of TMB substrate was added and the plates were incubated for 10 min at room temperature, after which 100 μL of stop solution was dispensed into each well. The optical density (OD) was read at 450 nm using an ELISA reader within 15 min. The Antibody Index Interpretation was as follows: less than (<) 0.9: no detectable antibody to toxoplasma IgG by ELISA; 0.9–1.1: borderline positive; >1.1: detectable antibody to toxoplasma IgG by ELISA.

### 2.7. Data Management and Analysis

Data collected were cross-checked for any errors. All of the questionnaires were given unique codes and the information was entered into the CSPro 7.6 data mask designed by the statistician. Continuous variables were reported as medians with 25th and 75th percentiles, and as means and standard deviations. Categorical variables were described as frequencies and percentages. The Wilcoxon rank sum test (Mann−Whitney U test) and Kruskal−Wallis rank sum tests were used for comparisons between the biomarker concentrations and TBI severity and outcomes. $p$-values < 0.05 were considered statistically significant.

### 2.8. Ethical Clearance and Administrative Authorizations

Ethical clearance for the study was obtained from the Institutional Review Board of the Faculty of Health Sciences (IRB-FHS), University of Buea; Reference N° 2022/1238-08/UB/IRB/FHS. Administrative authorization was obtained from the Laquantinie Hospital of Douala.

### 3. Results

#### 3.1. Characteristics of Participants

A total of 160 patients with a TBI of median age 32 (IQR26, 39) years were enrolled in the study between January 2021 and February 2022. Detailed clinical and demographic characteristics of the study cohort have previously been described (Buh et al., 2022, accepted). In brief, the median age of patients was 32 (IQR26, 39). Most patients were adolescents and young

adults aged 15–45 years (78%; 125), and 90% (144/160) of patients were males (144/160). Most participants (76%) had not finished secondary education. The median GCS was 12.0 (8.0–14.0). Driving after alcohol consumption was suspected in 33 (21%) of cases. Most of the patients (59%, 95) were referred directly (59%, 95) and 65 (41%) were secondary referrals. Mild TBI cases were the most common (41%; 66/160) presentation form, followed by moderate (34%; 55/160) and severe (24%; 39/160) TBI. CT scan was performed in 78% (125) cases and showed traumatic intracranial abnormalities in 64% (77/125) of cases. The two most common types of TBI were cerebral contusion (54%; 65/160) and extradural haemorrhage (49%; 59/160); neurosurgical intervention was carried out in 22% (17/77) of cases (Table 1).

**Table 1.** Sociodemographic and clinical characteristics.

| Characteristic | N (%) |
| --- | --- |
| **N** | **160** |
| **Age, Median (IQR) in years** | 32 (IQR26, 39) |
| **Gender** | |
| Female | 16 (10%) |
| Male | 144 (90%) |
| **Education** | |
| Graduate | 19 (12%) |
| No formal education | 9 (5.6%) |
| Matriculated | 7 (4.4%) |
| Not Known | 2 (1.2%) |
| Post graduate | 1 (0.6%) |
| Primary | 40 (25%) |
| Secondary | 82 (51%) |
| **Profession** | |
| Employee in service | 30 (19%) |
| Manual workers | 24 (15%) |
| Bike riders | 43 (27%) |
| Student | 16 (10%) |
| Unemployed | 21 (13%) |
| Others | 26 (16%) |
| **Marital status** | |
| Married | 67 (42%) |
| Not applicable | 6 (3.8%) |
| Single | 86 (54%) |
| Widowed | 1 (0.6%) |
| **Medico-social history** | |
| Diabetes | 2 (1.2%) |
| Hypertension | 14 (8.8%) |
| Smoking | 22 (14%) |
| Alcohol | 97 (61%) |
| **Influence of alcohol** | |
| None | 108 (68%) |
| Suspected | 33 (21%) |
| Unknown | 19 (12%) |
| **Symptoms of TBI** | |
| Loss of consciousness | 152 (95%) |
| Vomiting | 55 (34%) |
| Nausea | 21 (13%) |
| Ear bleed | 20 (12%) |
| Nasal bleed | 43 (27%) |
| Headache | 103 (64%) |
| Seizure | 7 (4.4%) |
| Agitation | 43 (27%) |

**Table 1.** *Cont.*

| Characteristic | N (%) |
|---|---|
| **N** | **160** |
| **Classification of TBI** | |
| Mild | 66 (41%) |
| Moderate | 55 (34%) |
| Severe | 39 (24%) |
| **Blood pressure** | N = 160 |
| Elevated | 15 (9.4%) |
| Hypertension | 62 (39%) |
| Hypotension | 19 (12%) |
| Normal | 64 (40%) |
| **Median Glasgow Coma Scale** | 12.0 (8.0, 14.0) |
| **Referrals** | |
| Direct | 95 (59%) |
| Indirect | 65 (41%) |
| **Complementary exams done to characterize injury** | N = 160 |
| CT Scan | 125 (78%) |
| If scan or MRI, traumatic abnormalities present | 77 (64%) |
| **Type of TBI** | N = 77 |
| Cerebral contusion | 25 (32%) |
| Extradural hematoma | 22 (29%) |
| Acute subdural haemorrhage | 18 (23%) |
| Intracerebral haemorrhage | 12 (16%) |
| Cerebral oedema | 10 (13%) |
| Meningeal haemorrhage | 8 (10%) |
| Mass effect pressure | 3 (3.9%) |
| Other type of TBI | 4 (5.2%) |
| **Neurosurgery** | N = 77 |
| Yes | 17 (22.1%) |
| No | 60 (78%) |

TBI: traumatic brain injury.

### 3.2. Serum Concentrations of Biomarkers and TBI Severity and Outcome

The levels of all biomarkers were significantly higher in acute traumatic brain injury patients compared with the controls (Table 2).

**Table 2.** Comparison of serum biomarker levels in TBI cases and controls.

| Biomarkers as Median (IQR) | TBI Cases N = 160 | Controls N = 15 | *p*-Value |
|---|---|---|---|
| S100B (pg/mL) | 28 (26, 33) | 0 (0, 0) | <0.001 |
| GFAP (pg/mL) | 1244 (277, 4042) | 13 (11, 21) | <0.001 |
| NFL (pg/mL) | 7 (2, 16) | 1 (1, 1) | <0.001 |
| Tau (pg/mL) | 1.15 (0.44, 2.87) | 0.32 (0.20, 0.49) | <0.001 |
| UCH-L1 (pg/mL) | 31 (12, 86) | 7 (4, 9) | <0.001 |

*p*-value is based on non-parametric comparison of median (IQR) (Wilcoxon rank sum test). UCHL-1: ubiquitin C-terminal hydrolase; NFL: neurofilament light; GFAP: glial fibrillary acidic protein; S100B: calcium binding protein. IQR: Interquartile range.

Serum concentrations of four of the TBI biomarkers generally increased with injury severity (Figure 1). The S100B levels did not demonstrate a clear increase in severe TBI compared with mild and moderate TBI. However, when the concentrations in the controls were compared with the severities, namely mild, moderate, or severe, there were significant differences ($p < 0.001$), as shown in Table 3.

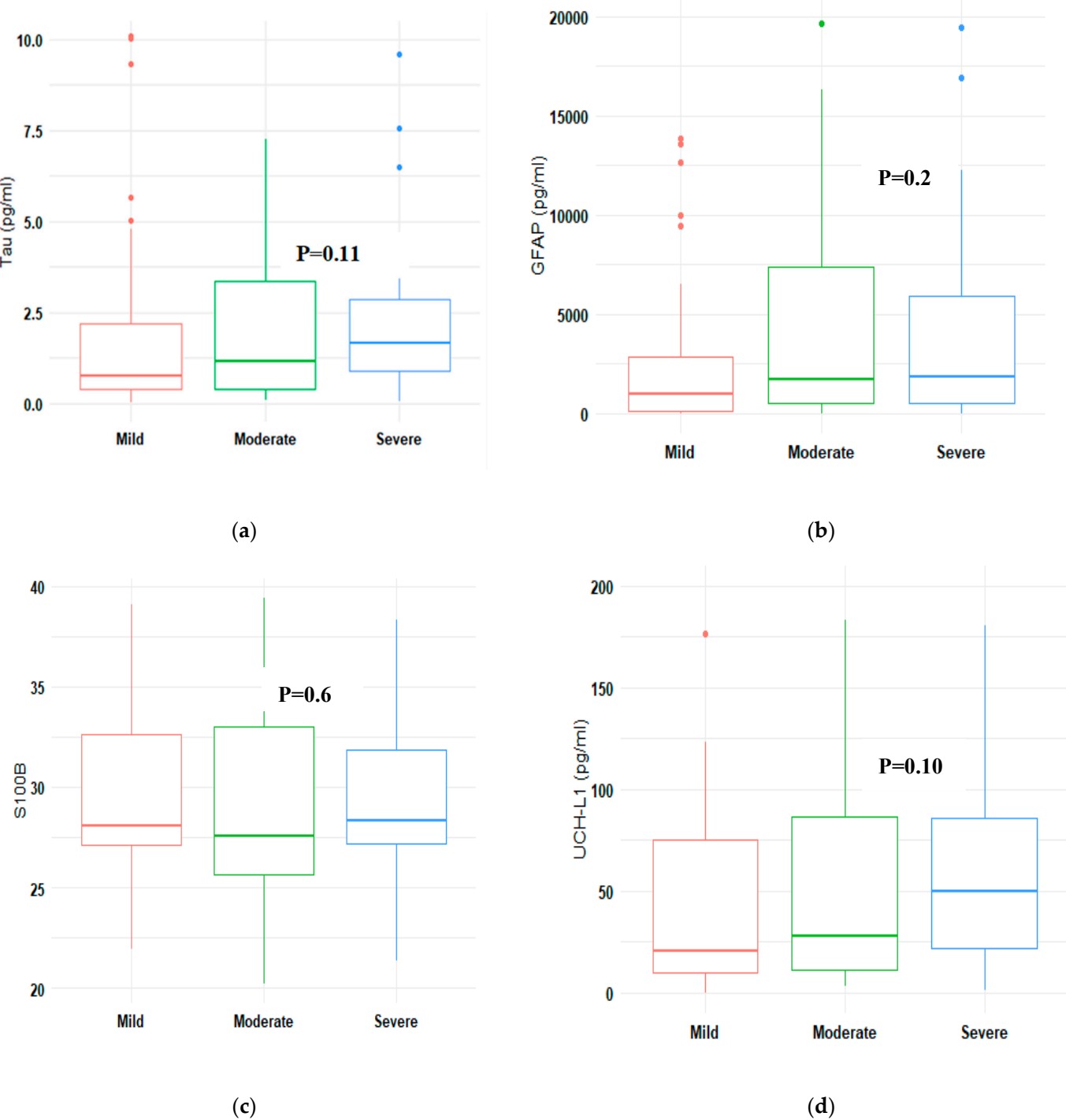

(**a**)

(**b**)

(**c**)

(**d**)

**Figure 1.** *Cont.*

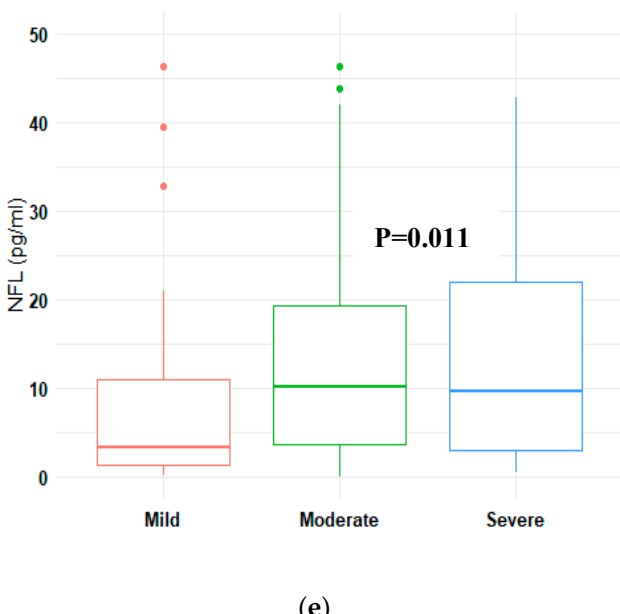

(**e**)

**Figure 1.** A comparison of median concentrations of biomarkers by TBI severity (**a**): Tau, (**b**) GFAP, (**c**) S100B, (**d**) UCH-L1, (**e**) NFL. UCH-L1: ubiquitin C-terminal hydrolase; NFL: neurofilament light; GFAP: glial fibrillary acidic protein; S100B: calcium binding protein; pg/Ml: picogram/millilitre.

**Table 3.** Concentration of neurobiomarkers according to TBI severity.

| Biomarkers as Median (IQR) | Controls | Traumatic Brain Injury Severity | | | *p*-Value |
|---|---|---|---|---|---|
| | | **Mild** | **Moderate** | **Severe** | |
| **N** | **15** | **66** | **55** | **39** | |
| S100B (pg/mL) | 0.0 (0, 0) | 28.1 (27.1, 32.6) | 27.6 (25.6, 33.0) | 28.3 (27.2, 31.9) | <0.001 |
| GFAP (pg/mL) | 13.0 (11, 21) | 1020 (147, 2831) | 1723 (503, 7.403) | 1876 (532, 5903) | <0.001 |
| NFL (pg/mL) | 1.0 (1, 1) | 3.0 (1, 11) | 10.0 (4, 19) | 10.0 (3, 22) | <0.001 |
| Tau (pg/mL) | 0.32 (0.20, 0.49) | 0.76 (0.39, 2.18) | 1.17 (0.40, 3.35) | 1.65 (0.88, 2.85) | <0.001 |
| UCH-L1 (pg/mL) | 7.0 (4, 9) | 20.0 (10, 75) | 28.0 (12, 87) | 50.0 (22, 86) | <0.001 |

*p*-value obtained by comparison of controls and mild TBI (Wilcoxon rank sum test; Wilcoxon rank sum exact test). UCHL-1: Ubiquitin C-terminal hydrolase, NFL: Neurofilament light, GFAP: Glial fibrillary acidic protein, S100B: Calcium binding protein, pg/mL: picogram/millilitre. IQR: Interquartile range.

Patients with an unfavourable outcome had higher levels of GFAP (*p* = 0.026) compared with those with a favourable outcome. No significant differences between patients with favourable and unfavourable outcomes were found for the other biomarkers considered, although a clear trend towards higher values of NFL in patients with an unfavourable outcome was demonstrated. The total tau levels were significantly higher in patients who died from injury compared with the survivors (*p* = 0.017). Except for S100B, the levels of other biomarkers (GFAP, NFL, and UCH-L1) were higher in non-survivors, but these differences did not reach statistical significance (Table 4).

GFAP and total tau levels were significantly higher in CT positive compared with CT negative patients (GFAP: *p* = 0.004; NFL: *p* = 0.002). UCHL-1 was also found to be sensitive as the values were relatively high in those who showed traumatic intracranial abnormalities with the CT scan (21 pg/mL for negative CT-scan vs. 42 pg/mL for positive CT-scan), although this increase was not statistically significant. However, when restricting the analysis to patients with mild TBI (*n* = 42), no significant difference between CT-positive and CT-negative patients was found. Significant differences for GFAP (*p* = 0.038) and NFL (*p* = 0.005) remained when mild and moderate TBI cases (n = 86) were combined (Table 5).

**Table 4.** Neurobiomarkers according to GOSE.

| Biomarkers as Median (IQR) | Survival N= 130 | Death, N = 22 | Favorable N = 114 | Unfavorable N = 38 | *p*-Value (Favourable vs. Unfavourable GOSE | *p*-Value (Death vs. Survival) |
|---|---|---|---|---|---|---|
| S100B (pg/mL) | 28.1 (26.3, 32.6) | 28.2 (27.2, 33.7) | 28.1 (26.3, 32.6) | 28.0 (26.4, 32.1) | >0.9 | 0.4 |
| GFAP (pg/mL) | 1161 (219, 3349) | 2811 (569, 8611) | 1100 (132, 3310) | 2545 (582, 7356) | 0.026 | 0.087 |
| NFL (pg/mL) | 7.0 (2, 15) | 12.0 (2, 27) | 5.0 (2, 15) | 10.0 (5, 26) | 0.11 | 0.3 |
| Tau (pg/mL) | 0.98 (0.40, 2.31) | 2.84 (1.03, 3.88) | 1.15 (0.40, 2.62) | 1.55 (0.63, 3.44) | 0.3 | 0.017 |
| UCH-L1 (pg/mL) | 24.0 (12, 83) | 53.0 (20, 78) | 30.0 (12, 88) | 40.0 (13, 83) | 0.6 | 0.2 |

Median (IQR); Wilcoxon rank sum test. All TBI cases are included. Favourable: Glasgow Outcome Scale Extended (GOSE) 5–8; Unfavourable; GOSE 1–4. UCHL-1: ubiquitin C-terminal hydrolase; NFL: neurofilament light; GFAP: glial fibrillary acidic protein; S100B: calcium binding protein; pg/mL: picogram/millilitre. IQR: Interquartile range.

**Table 5.** Biomarker concentrations and CT-scan outcome.

| Biomarkers as Median (IQR) | CT Negative N = 44 | CT Positive N = 77 | *p*-Value (All Severities; Mild, Moderate, Severe) | *p*-Value (Only Mild TBI) | *p*-Value (Mild and Moderate) |
|---|---|---|---|---|---|
| S100B (pg/mL) | 28.3 (26.3, 33.0) | 28.0 (26.4, 32.5) | >0.9 | 0.6 | 0.8 |
| GFAP (pg/mL) | 928 (146, 1955) | 1809 (475, 7130) | 0.004 | >0.9 | 0.038 |
| NFL (pg/mL) | 3.0 (2, 10) | 10.0 (4, 22) | 0.002 | 0.2 | 0.005 |
| Tau (pg/mL) | 0.95 (0.39, 1.96) | 1.35 (0.46, 3.14) | 0.2 | 0.4 | 0.7 |
| UCH-L1 (pg/mL) | 21.0 (9, 48) | 42.0 (15, 86) | 0.085 | 0.9 | 0.076 |

Median (IQR); Wilcoxon rank sum test. All TBI cases were included. UCHL-1: ubiquitin C-terminal hydrolase; NFL: neurofilament light; GFAP: glial fibrillary acidic protein; S100B: calcium binding protein; pg/mL: picogram/millilitre. IQR: Interquartile range.

### 3.3. Toxoplasma gondii Infection and Biomarkers Concentrations

*Toxoplasma gondii* infection was recorded in 33% (52/160) of TBI cases. The median age for *T. gondii* positive patients was 30 (IQR23, IQR39) years and there were no significant differences between *T. gondii*-positive and negative TBI patients in terms of TBI severity (*p* = 0.7), as shown in Supplemental Table S1. When the biomarker concentrations between *T. gondii* negative vs. positive cases were compared, only S100B was found to be significantly higher in *T. gondii*-positive TBI cases (Table 6).

**Table 6.** Influence of toxoplasma status on biomarkers.

| Biomarker as Median (IQR) | Toxoplasma Negative, N = 108 [1] | Toxoplasma Positive, N = 52 [1] | *p*-Value [2] |
|---|---|---|---|
| S100B (pg/mL) | 27.2 (24.9, 28.1) | 36.2 (32.6, 37.9) | <0.001 |
| GFAP (pg/mL) | 1651 (351, 5394) | 839 (244, 3360) | 0.4 |
| NFL (pg/mL) | 7.0 (2, 15) | 7.0 (2, 21) | >0.9 |
| Tau (pg/mL) | 1.39 (0.53, 3.39) | 0.82 (0.38, 2.12) | 0.10 |
| UCH-L1 (pg/mL) | 33.0 (13, 94) | 28.0 (10, 68) | 0.2 |

[1] Median (IQR), [2] Wilcoxon rank sum test. IQR: Interquartile range.

### 4. Discussion

The development of clinically validated traumatic brain injury biomarker tests can improve treatment approaches and prognostic estimates in patients with TBI in acute care settings. In what is, to the best of our knowledge, the first TBI biomarker study in Cameroon and SSA, we report how five biomarker concentrations vary with TBI severity, CT scan positivity, and outcome, and explore the influence of *Toxoplasma gondii* seropositivity on the biomarker levels.

### 4.1. Concentrations of Serum Biomarkers and the Association with Traumatic Brain Injury Severity

The serum concentrations of the TBI biomarkers studied (S100B, NFL, UCH-L1, Tau, and GFAP) were significantly higher compared with the controls. These proteins are mainly found in CNS and can be found in blood and CSF in traumatic brain injury or other disruptions in CNS [14,16,21]. The serum concentrations of the TBI biomarkers considered in this study generally increased with TBI severity (mild to severe), except for S100B, which increased slightly in severe TBI, but not in moderate TBI, similar to reports in the literature [3,12,22,31,32]. There was a significant increase in the serum levels of NFL, while the serum levels of S100B remained almost the same from mild to severe TBI. This is in line with reports by Shahim et al. [33], where NFL was found to reflect TBI severity after traumatic brain injury. However, our results, in part, do not corroborate those of the above cited studies, as most of the biomarkers in the above studies showed significant variations from mild to moderate and severe TBI. Differences in sample sizes and study setting may be accountable for this. The high level of biomarker concentration in non-survivors may be explained by neuronal and glial damage [34]. Moreover, biomarker levels reflect damage at the cellular level, which may not be seen with imaging techniques [15], and thus motivates further biomarker research in SSA to support their role in clinical decision making.

### 4.2. Concentrations of the Biomarkers and 6 Months Outcome

We found significantly higher levels of t-Tau in non-survivors compared with survivors, which is in line with the results obtained by Wang et al. [35], where increased levels of tau were found to be associated with poorer outcomes after TBI. It was also consistent with findings by Korley et al. [24], where higher values of tau and UCH-L1 predicted mortality. However, in contrast with other studies, we did not find significant differences in the levels of other biomarkers considered (GFAP, NFL, S100B, and UCH-L1) between non-survivors and survivors. Nevertheless, except for S100B, the levels of other biomarkers were higher in non-survivors. This corroborates studies reporting higher values of these biomarkers in non-survivors [23,24,34,36]. Regarding the comparison between favourable vs. unfavourable outcomes, GFAP was significantly higher in patients with an unfavourable outcome 6 months after TBI, in line with the results reported by Korley et al. [24] and Helmrich et al. [23]. These studies, however, also reported significant associations with outcome for other biomarkers, which were not demonstrated in our findings. This may be explained by our relatively small sample size (160) compared with the over 2000 participants considered in the studies mentioned above, as well as to differences in case-mix and time of sampling.

### 4.3. Concentrations of the Biomarkers and CT Positive and Negative Scans

The correlation of the biomarker concentrations between the TBI-positive CT scan (77%) vs. TBI-negative CT scan (23%) showed that two of the five biomarkers (GFAP and NFL) were significantly associated with CT positivity after traumatic brain injury. This result is partly in line with several studies [3,21,22,37], where the sensitivity of GFAP and UCH-L1 were demonstrated. Furthermore, McMahon et al. [38] demonstrated that GFAP biomarker testing in emergency services could eliminate unnecessary CT scans in 12 to 30% of TBI patients. However, when the comparison was made considering mild TBI alone, no significant differences were obtained between CT positive and CT negative scans with any of the tested biomarkers. This observation is not consistent with reports by Czeiter et al. [22], where they reported an incremental diagnostic value for GFAP in mild TBI cases. The difference in results could be accounted for by our relatively low event rate in mild TBI ($n = 14$) and by the differences in study settings.

Although our sample size was too small to draw definitive conclusions, the significantly higher levels of NFL in CT-positive cases may motivate further research into the role of this biomarker in predicting the presence of CT abnormalities with a focus on the timing of sampling.

The use of TBI biomarker testing for informing the need for CT scanning is particularly relevant in resource-limited settings as the price of CT scans and MRI are high. More than half of the population of Cameroon (60%) cannot access appropriate healthcare because of the high costs and 70% spend out-of-pocket for their healthcare as no universal healthcare services are yet offered [39]. Therefore, the development and implementation of biomarkers for ruling out CT scans when not necessary is promising and could reduce the health expenditure of the patients and proxies in mild TBI, and thus improve healthcare provision and health.

### 4.4. Concentrations of the Biomarkers according to Toxoplasma gondii Infection

We hypothesized that a latent *T. gondii* infection might influence the neuropathology in traumatic brain injury. Recent literature attests that latent infections with *T. gondii* may cause neuro-inflammation, which could worsen the neuro-inflammatory component of TBI [40]. However, it has not yet been experimentally established how *T. gondii* affects the brain in TBI. We therefore aimed to investigate if TBI patients infected with *T. gondii* would show higher levels or not of serum TBI biomarkers.

*Toxoplasma gondii* positivity was observed in 33% of cases, with a median age of 30 (23, 39) years, similar to reports in the literature where one third of the world population is said to be infected with *T. gondii* [40–42]. S100B levels were higher in *T. gondii*-positive patients. To the best of our knowledge, this is the first study exploring the effects of *T. gondii* infection on biomarkers in TBI. However, Ayyildiz et al. [43] studied S100B serum levels associated with *T. gondii* positivity in Alzheimer disease, and found no significant variation in the levels of serum S100B between *T. gondii*-positive and -negative Alzheimer patients. This variation in results may be due to the fact that their sample size was small (33) or simply to a difference in the underlying pathology, e.g., chronic versus acute. The authors suggest that the use of different genotypes of *T. gondii* in future studies may add to literature. As of today, it is not clear if TBI outcomes may differ in individuals infected with neurotrophic parasites such as *T. gondii* and how seropositivity to *T. gondii* in traumatic brain injury influences biomarker concentrations. The possibility that elevated S100B levels may have had an extracerebral origin cannot be excluded. However, the potential elevation of serum levels of S100B in neuro-inflammation after TBI should not be overlooked, but instead, further studies with larger sample sizes and more accurate laboratory methods may help to clarify the possible effects of *T. gondii* on S100B serum or plasma concentrations. Furthermore, experimental studies have shown that *T. gondii* infection reduces cerebral microvascular perfusion and induces neuro-inflammation through activation of the cerebral endothelial cells [40,41]. These results support the concept that latent neuro-parasitic disease, like *T. gondii* infection, might aggravate the disease process in acute neurological disorders, which could be reflected in biomarker concentrations. Currently, work is ongoing at the Monash University in Australia to determine the effects of *T. gondii* infection on acquired brain injury (TBI and stroke) and its outcome [44]. Further studies considering larger sample sizes are needed to elucidate the effects of latent neuroparasitic disease on the pathophysiology and outcome of TBI. This would contribute to the knowledge of the pathomechanisms after TBI, and thus the diagnosis and care of TBI patients, particularly in low-resource settings.

## 5. Strengths and Limitations

This is the first study designed considering TBI biomarkers in Cameroon and their association with TBI severity, CT scan positivity/negativity, and 6-month outcome with GOSE, as well as the influence of concurrent *Toxoplasma gondii* infection on serum biomarker levels. This could serve as baseline information for future research on TBI biomarkers in Cameroon and other parts of Africa, as well as on *T. gondii* neuropathology in acute TBI. This study also reported the potential of NFL as a sensitive marker of CT abnormalities, which will subsequently need to be studied further for more accurate findings. As of now, this characteristic is noted with two of the TBI biomarkers: GFAP and UCH-L1. We wish to carry out similar studies in the future with larger sample sizes and considering more

trauma centres in Cameroon. A limitation of our study was the relatively low sample size, and that CT scans could not be obtained in all patients. In addition, our study was limited by the fact that we did not record the different timelines at which the blood samples were collected within the 24 h of injury. Another limitation was loss to follow-up of a few cases (8) at the 6-month evaluation due to inaccurate contact information. Despite these limitations, our results are pertinent, offering insights regarding TBI biomarker testing in LMIC, as well as determining a possible influence of *T. gondii* seropositivity on the TBI biomarker concentrations.

## 6. Conclusions and Implications

The serum concentrations of the five TBI biomarkers considered in this study generally increased from mild to severe TBI, although this increase was statistically significant only for NFL. However, when the concentrations in the controls were compared with any of the severities, namely mild, moderate, or severe, there were significant differences. We report that GFAP was associated with an unfavourable outcome 6 months after TBI. The concentrations of tested biomarkers were generally increased in non-survivors, although the increase was significant for only one of these biomarkers (Tau). Two (GFAP and NFL) of the five biomarkers were found to predict CT abnormalities. When the biomarker concentrations between *T. gondii* negative vs. positive cases were compared, S100B was significantly higher in *T. gondii* positive TBI cases. Future studies should be conducted with larger sample sizes and recruiting more trauma centres around Cameroon and SSA in order to draw more definite conclusions on the use of biomarkers as TBI diagnostic and prognostic tools in resource-limited settings, as well as on *Toxoplasma gondii* neuropathology in traumatic brain injury. NFL could be studied further to explore its sensitivity towards CT-positive or negative scans. Finally, the governments of SSA countries should promote and encourage research in this area, which is promising in the prognosis and care of TBI in SSA.

**Supplementary Materials:** The following supporting information can be downloaded at: https://www.mdpi.com/article/10.3390/neurosci4030015/s1. Supplemental Table S1 shows that *Toxoplasma gondii* infection was recorded in 33% (52/160) of TBI cases. The median age for *T. gondii* positive patients was 30 (IQR23, IQR39) years and there were no significant differences between *T. gondii* positive and negative TBI patients in terms of TBI severity ($p = 0.7$).

**Author Contributions:** F.C.B.: study design, data collection, and writing. G.S.T.: study design and editing. F.H.K.: laboratory analysis and editing. K.W.W.: laboratory analysis and editing. A.I.R.M.: study design and editing. M.M.: study design and editing. B.K.M.: editing. E.Y.: data analysis. P.J.A.H.: study design and editing. I.U.N.S.: study design, editing, and validation. All authors have read and agreed to the published version of the manuscript.

**Funding:** The study received no external funding.

**Institutional Review Board Statement:** The study was conducted according to the guidelines of the Declaration of Helsinki and was approved by the Institutional Review Board of the Faculty of Health Sciences, University of Buea, Cameroon (IRB-FHS). Reference number: 2022/1238-08/UB/SG/IRB/FHS.

**Informed Consent Statement:** Informed consent was obtained from all subjects involved in the study.

**Data Availability Statement:** Not applicable.

**Acknowledgments:** We acknowledge the administration and health personnel of the Laquintinie Hospital Douala for facilitating the research process. Our sincere thanks to the Centre of Neuroproteomics and Biomarker Research, University of Florida (USA), for their support in the shipment of blood samples and analyses of the four TBI biomarkers. We also acknowledge the support of the NIHR Global Health Research Group on Acquired Brain and Spine Injury, Cambridge University, (UK) for their support in the realization of this study.

**Conflicts of Interest:** The authors declare no conflict of interest.

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
