# Peer review of "Serum Biomarker Concentrations upon Admission in Acute Traumatic Brain Injury: Associations with TBI Severity, Toxoplasma gondii Infection, and Outcome in a Referral Hospital Setting in Cameroon"

_neurosci, doi:10.3390/neurosci4030015_

Round 1

Reviewer 1 Report

The authors collected blood samples from TBI patients in Cameroon. This is meaning because it is an understudied disease in an understudied country. The blood samples were tested for TBI biomarkers (GFAP, NFL, Tau, UCHL1 & S100B) as well as T Gondii IgM. They found that compared to controls, TBI increased all biomarkers. In addition, GFAP was associated with unfavorable GOSE at 6-mo and tau associated with non-survival. GFAP & NFL were associated with CT positivity.

Here are some of my major comments:

- P.3 Lin 132

It is known that blood coagulation interferes with tau concentration. Plasma collection is preferred for analysis of blood tau level. The authors may want to comment on this in the discussion.

- P.3 Line 142

If possible, please show a more precise timing of sample collection, in terms of hours (e.g. median & IQR).

- P.4  Section 2.6

Why  does the authors choose to detect T Gondii IgM ab, if their hypothesis is T Gondii latent infection may exacerbate TBI biomarkers? IgG would be more appropriate.

- P. 7 Line 225 + P9. Line 281

"Serum concentrations all TBI biomarkers generally increased with injury severity". This is not true. S100B did not increase with more severe TBI.

- P.7 Table 3

Please explain the statistical method used in comparison of "TBI severity". (The first p-value)

- P.8 Table 4

Please define "favourable" & "unfavourable" outcomes

P.7 Table 3, P.8 Table 4, Table 5

It appears that in this study, S100B cannot differential TBI severity, survival, GOSE outcome, or CT +/-, though it does show significant difference between control vs TBI.

This makes one wonder if the S100B assay is fully validated? Are there evidence that the test does not have sensitivity/signal saturation issues? Or dilution linearity issues?

P.10 Line 335 + Conclusion

The association of serum positivity for T.gondii and higher TBI biomarker should be interpreted with caution:

1. T Gondii IgM is used for this study. So the serum positivity show here is an evidence of new infection, not latent infection.

(Line 335: "We hypothesized that a latent T. gondii infection might influence the neuropathology 335 in traumatic brain injury.")

2. If the authors want to test the hypothesis of latent infection, IgG is more suitable.

3. Only S100B is shown to be higher in T gondii positive but not T gondii negative (Table 6), and not other markers (GFAP, NFL, Tau, UCHL1). In fact, GFAP, Tau and UCHL1 are showing a trend of decrease (non-significant).

Co-incidentally, S100B did not differentiate TBI survival, CT positivity nor TBI severity. This makes me wonder if this effect of T gondii on S100B is due to random chance.

Keep in mind that throughout this study, a statistical alpha of 0.05 is used and that there are 5 blood biomarkers. Comparisons have been made for many outcomes (TBI severity, survival, CT positivity, GOSE, etc) Due to the many comparisons, there could be a chance of reaching p-value <0.05 just due to random chance alone (unless there were methods that adjusted for multiple comparison)

For these reasons, I'd suggest the authors interpret the findings of the potential effects of T Gondii with much caution, unless there is other validation experiment, e.g. either by validating the results in an independent cohort, or recruiting more TBI/control patients for T gondii antibody measurements. Alternatively, the authors may also try measuring T Gondii DNA or IgG in blood, and see if the effects on TBI biomarkers (S100B) persists.

Minor comments:

- P.2 Line 65

UCH-L1 and GFAP. May be more appropriate to clarify at first mention that FDA approved the use of the 2 biomarkers for evaluating if TBI patients need a CT scan.

- Line 93

Please clarify: 09 hectares or 90 hectares?

- Data presentation

The authors reported the data using tables (reporting median, IQR, and p-value). It would be more clear if they can plot some of the major key findings with plots showing each data point. This can help interpret data distribution and if there are outliers.

Author Response

Please find the responses in the attached file.

Reviewer 2 Report

In this paper, the authors introduce the associations in acute TBI between serum biomarker concentrations and TBI severity, CT-scan findings and outcome, explore the possible influence of Toxoplasma gondii infection. Overall, it’s a good paper.

However, I have a few concerns regarding the manuscript:

1. How were the concentrations of these markers detected? Can you provide more details on the statistical methods employed?

2. Does the number of cases in this study provide sufficient evidence to conclude that biomarkers like GFAP and NFL can predict TBI CT abnormalities? Would it be beneficial to include control cases for comparison?

3. On Page 11, the phrase "the first study" is used multiple times. It would be preferable to use alternative wording.

4. Some sentences in the manuscript could be rephrased for better clarity and to eliminate redundancy. It would be beneficial to revise these sentences accordingly.

Some sentences in the manuscript could be rephrased for better clarity and to eliminate redundancy. It would be beneficial to revise these sentences accordingly.

Author Response

(The authors gave the same response as above.)

Round 2

Reviewer 1 Report

The authors have made satisfactory edits to the text/added additional comments and graphs.

I have only 2 remaining minor comments:

1. Abstract:

The last sentence in abstract is "S100B levels were significantly higher in T. gondii seropositive TBI patients, compared to seronegative patients." This seems to be redundant to a previous sentence in the same paragraph "S100B levels were higher (P<0.001) in TBI patients seropositive to Toxoplasma gondii."

2. Regarding Table 4: Neurobiomarkers according to the GOSE.

There were Survival (N= 130), Death (N = 22), Favorable (N = 122), and Unfavorable (N = 38).

However, it appears that the total survival + death (130 + 22) does not add up to total number of TBI patients (160).

And the total number of favorable + unfavorable does not add up to survival. 

Please clarify.

Author Response

(The authors gave the same response as above.)
